

# Assessing uncertainties of a geophysical approach to estimate surface fine particulate matter distributions from satellite observed aerosol optical depth

5    Xiaomeng Jin[1], Arlene M. Fiore[1], Gabriele Curci[2,3], Alexei Lyapustin[4], Kevin Civerolo[5], Michael Ku[5], Aaron van Donkelaar[6], Randall V. Martin[6,7]

1. Department of Earth and Environmental Sciences of Lamont-Doherty Earth Observatory and Columbia University, Palisades, NY, USA

2. Department of Physical and Chemical Sciences, University of L'Aquila, L'Aquila, Italy

10    3. CETEMPS, University of L'Aquila, L'Aquila, Italy

4. NASA Goddard Space Flight Center, MD, USA

5. New York State Department of Environmental Conservation, Albany, NY, USA

6. Department of Physics and Atmospheric Science, Dalhousie University, NS, Canada

7. Smithsonian Astrophysical Observatory, Harvard-Smithsonian Center for Astrophysics, 15    Cambridge, MA, USA

Corresponding Author's Email Address: xjin@ldeo.columbia.edu



## Abstract

Health impact analyses are increasingly tapping the broad spatial coverage of satellite aerosol optical depth (AOD) products to estimate human exposure to fine particulate matter ($PM_{2.5}$). We use a forward geophysical approach to derive ground-level $PM_{2.5}$ distributions from satellite AOD

at 1 $km^2$ resolution for 2011 over the Northeast USA by applying relationships between surface $PM_{2.5}$ and column AOD (calculated offline from speciated mass distributions) from a regional air quality model (CMAQ; $12 \times 12$ $km^2$ horizontal resolution). Seasonal average satellite-derived $PM_{2.5}$ reveals more spatial detail and best captures observed surface $PM_{2.5}$ levels during summer. At the daily scale, however, satellite-derived $PM_{2.5}$ is not only subject to measurement

uncertainties from satellite instruments, but more importantly, to uncertainties in the relationship between surface $PM_{2.5}$ and column AOD. Using 11 ground-based AOD measurements within 10 km of surface $PM_{2.5}$ monitors, we show that uncertainties in modeled $PM_{2.5}$/AOD can explain more than 70% of the spatial and temporal variance in the total uncertainty in daily satellite-derived $PM_{2.5}$ evaluated at $PM_{2.5}$ monitors. This finding implies that a successful geophysical approach to

deriving daily $PM_{2.5}$ from satellite AOD requires model skill at capturing day-to-day variations in $PM_{2.5}$/AOD relationships. Overall, we estimate that uncertainties in the modeled $PM_{2.5}$/AOD lead to an error of 11 $\mu g/m^3$ in daily satellite-derived $PM_{2.5}$, and uncertainties in satellite AOD lead to an error of 8 $\mu g/m^3$. Using multi-platform ground, airborne and radiosonde measurements, we show that uncertainties of modeled $PM_{2.5}$/AOD are mainly driven by model uncertainties in aerosol

column mass and speciation, while model uncertainties of relative humidity and aerosol vertical profile shape contribute some systematic biases. The parameterization of aerosol optical properties, which determines the mass-extinction efficiency, also contributes to random uncertainty, with the size distribution the largest source of uncertainty, and hygroscopicity of inorganic salt the second.



Future efforts to reduce uncertainty in geophysical approaches to derive surface $PM_{2.5}$ from satellite AOD would thus benefit from improving model representation of aerosol vertical distribution and aerosol optical properties, to narrow uncertainty in satellite-derived $PM_{2.5}$.

*Keywords*: $PM_{2.5}$, aerosol optical depth, CMAQ, DISCOVER-AQ, aerosol size distribution,

5    aerosol hygroscopic growth.



## 1    Introduction

Exposure to ambient fine particulate matter ($PM_{2.5}$) is estimated to cause more than 4 million attributable deaths worldwide in 2015 (Lim et al., 2016), and is associated with an increase in the risk of cardiovascular and respiratory disease (Dominici et al., 2006; Peng et al., 2009).

Evidence is emerging that exposure to $PM_{2.5}$ has adverse health effects even at low concentrations (Crouse et al., 2012; Shi et al., 2015). Early studies relied on the nearest ground-based monitors to estimate $PM_{2.5}$ exposure (e.g. Dockery et al., 1993; Laden et al., 2006), but lack of resolution of spatial and temporal gradients in population exposure may lead to substantial errors in health impact analyses.

Satellite remote sensing, which fills a spatial gap in ground-based networks, is playing an increasingly important role in $PM_{2.5}$ exposure assessment (Cohen et al., 2017; Jerrett et al., 2017). Aerosol optical depth (AOD), a measure of the sum of light extinction by aerosols within the atmospheric column, is retrieved from a number of satellite instruments. The Moderate Resolution Imaging Spectroradiometer (MODIS) on board Terra and Aqua has provided twice-daily global

AOD data for nearly two decades, and the Multi-Angle Implementation of Atmospheric Correction (MAIAC) product has refined the spatial resolution retrieved from MODIS to 1 km (Lyapustin et al., 2011; 2012), offering the potential to reveal aerosol spatial variability within urban cores (Hu et al., 2014). A big challenge to inferring near-surface $PM_{2.5}$ from column AOD retrieved from satellite instruments is to describe accurately the non-linear and spatiotemporally varying

relationship between $PM_{2.5}$ and AOD, which depends on aerosol chemical composition, vertical profiles, aerosol optical properties and the ambient environment (Griffin et al., 2012). Approaches to link satellite AOD with $PM_{2.5}$ exposures are often classified into two categories: statistical (e.g. Di et al., 2016; Hu et al., 2014; Kloog et al., 2014) and geophysical (e.g. van Donkelaar et al.,



2010; 2006). A two-stage process is also used with a geophysical approach followed by a statistical approach (e.g. van Donkelaar et al., 2015; de Hoogh et al., 2016; Shaddick et al., 2017).

Statistical approaches fit an optimized relationship between ground-based $PM_{2.5}$ and satellite AOD along with other predictors (e.g. land use, meteorology, traffic density etc.) using

methods such as multiple linear regression (e.g. Gupta and Christopher, 2009; Lee et al., 2016), geographic regression (Hu et al., 2014), generalized additive models (e.g. Kloog et al., 2014), or machine learning (Di et al., 2016). In regions with high monitor density, the statistical methods generally agree better with ground-based observations than $PM_{2.5}$ derived with geophysical approach, but statistical methods rely on the availability of ground-based monitors to train the

statistical model, and are thus limited to regions with dense monitoring networks.

The geophysical approach that has been applied to AOD is a process-based forward approach that uses chemical transport models to explicitly simulate the spatially and temporally varying relationship between column AOD and $PM_{2.5}$ (van Donkelaar et al., 2006). The satellite-derived $PM_{2.5}$ is calculated by taking the product of satellite AOD with the modeled ratio of $PM_{2.5}$

to AOD (van Donkelaar et al., 2006):

$$PM_{2.5\_sat} = AOD_{sat} \times \frac{PM_{2.5\_model}}{AOD_{model}} \qquad (1)$$

This geophysical approach has the advantage of broad spatial coverage that is not limited by the availability of *in-situ* measurements (van Donkelaar et al., 2006), and thus has been integral for studying the global burden of disease attributable to ambient air pollution (Cohen et al., 2017).

Van Donkelaar et al. (2010) estimate global annual average $PM_{2.5}$ using AOD observed from both MODIS and MISR (Multiangle Imaging Spectroradiometer) by $PM_{2.5}$-AOD relationships from a global chemical transport model (GEOS-Chem). They estimate an overall uncertainty of around



25% for annual average satellite-derived $PM_{2.5}$, but the uncertainty of the geophysical approach at short-time scale is expected to be larger (van Donkelaar et al., 2012).

The overall uncertainty in deriving surface $PM_{2.5}$ with the geophysical approach consists of uncertainty in the satellite AOD as well as the modeled $PM_{2.5}/AOD$. First, satellite observations

of AOD are subject to uncertainties due to the viewing geometry, the presence of clouds and snow, and choices involved in modeling optical aerosol and surface properties (Superczynski et al., 2017; Toth et al., 2014). Second, since the relationship between $PM_{2.5}$ and AOD is non-linear and multivariate, modeled $PM_{2.5}/AOD$ is subject to model uncertainties in aerosol vertical distributions, aerosol speciation and the ambient environment. Third, even if a model simulates accurately the

aerosol mass distribution, calculating AOD in models generally requires assumptions regarding the aerosol size distribution, aerosol species density, refractive index and hygroscopic growth factors, all of which are sources of uncertainties (Curci et al., 2015). The ability of a particle to scatter and absorb light largely depends on its size, which varies significantly in nature (Stanier et al., 2004). As resolving the size distribution is computationally expensive (Adams, 2002), aerosols

are typically assumed to follow a certain distribution (e.g. log-normal), which can introduce error. Moreover, aerosol water uptake (hygroscopicity) affects the aerosol size and optical properties, but the representation of hygroscopic factors in models varies considerably by study (Chin et al., 2002; Curci et al., 2015; Drury et al., 2010). The hygroscopic growth factor for organic carbon (OC) is especially uncertain, varying considerably by organic species, and is poorly represented in models

(Ming et al., 2005; Jimenez et al., 2009; Latimer and Martin, 2018). The impacts of these uncertainties on aerosol radiative forcing have been studied extensively in literature, but their impacts on deriving surface $PM_{2.5}$ from satellite-based column AOD have not yet been quantified.



Here, we estimate PM$_{2.5}$ distributions over the Northeast USA for 2011 using a geophysical approach that combines MAIAC AOD data with modeled PM$_{2.5}$/AOD relationships simulated with a regional air quality model (CMAQ). Compared to the global GEOS-Chem model used by van Donkelaar et al. (2016), CMAQ has finer spatial resolution ($12 \times 12$ km$^2$) and a locally refined emission inventory (see Sect. 2.2). We use an ensemble of surface (AQS, AERONET, IMPROVE, CSN), aircraft (DISCOVER-AQ) and radiosonde (IGRA) measurements to evaluate different sources of uncertainties in satellite-derived PM$_{2.5}$, especially at daily scale. To evaluate the sensitivities of satellite-derived PM$_{2.5}$ to the parameterization of aerosol optical properties, we conduct a series of sensitivity tests in an offline AOD calculation package (FlexAOD). The overarching goal of the comprehensive uncertainty analysis is to assess the relative importance of each uncertain factor, thereby advancing the process-level understanding of the relationship between satellite AOD and surface PM$_{2.5}$ air quality.

## 2  Data and methods

### 2.1  Satellite AOD

We use the high-resolution (1 km) daily AOD products retrieved from the MODIS instruments onboard Terra and Aqua satellites with the new MAIAC algorithm that is based on time series analysis and image processing techniques (Lyapustin et al., 2011; 2018). The spatial resolution of MAIAC (1 km) is finer than the conventional MODIS Dark Target and Deep Blue AOD products. The MAIAC algorithm improved upon the earlier Dark Target retrieval algorithm (MOD04) by explicitly including bi-directional reflectance (rather than the parameterized Dark Target approach), which improves accuracy over brighter surfaces, with similar accuracy over dark and vegetated surfaces (Lyapustin et al., 2011).





Using the quality flags provided, we filtered out pixels with or adjacent to cloud, snow or ice. We follow the approach of Hu et al. (2014) to combine daily MAIAC AOD from Terra (overpasses around 10:30 AM local time) and Aqua (overpasses around 1:30 PM local time). For the pixels where both Terra and Aqua have valid data, we take the average to reflect the mean

daytime AOD. For pixels where only one instrument has valid data, AOD may be biased accordingly towards morning or afternoon conditions. We find, on average, Terra-MAIAC AOD is higher than Aqua-MAIAC AOD by 0.005 (about 5% of the annual average AOD) over the Northeast USA in 2011, reflecting diurnal variations of AOD (Green et al., 2012) and potential calibration differences (Levy et al., 2018). To account for these differences, we fit two linear

equations (R = 0.87) between Terra-MAIAC ($AOD_T$) and Aqua-MAIAC AOD ($AOD_A$):

$$\widehat{AOD}_T = 0.84 AOD_A + 0.019 \qquad (2)$$

$$\widehat{AOD}_A = 0.88 AOD_T + 0.005 \qquad (3)$$

We use Eq. (2) and (3) to predict the AOD from the other instrument when one of them is missing, and then take the average. We find little seasonal variation in the linear relationship.

**2.2  CMAQ model**

The Community Multiscale Air Quality Modeling System (CMAQ) is a regional multipollutant air quality model developed and maintained by the U.S. Environmental Protection Agency (EPA). We use the CMAQ (v5.0.2) model simulations for 2011 conducted at New York State Department of Environmental Conservation (NYSDEC) for air quality planning purposes.

The simulations are conducted for the eastern USA with 12 km horizontal resolution and 35 vertical layers extending up to 50 hPa. The meteorological fields to drive CMAQ are provided by annual Weather Research and Forecast (WRF) v3.4 model simulations over continental United States. Chemical boundary conditions are from the GEOS-Chem ($2° \times 2.5°$) global chemical





transport model (Bey et al., 2001, version 8) generated by EPA. The emission inventory is based

on the 2011 National Emission Inventory (NEI) and processed through the Sparse Matrix Operator

Kernel Emissions (SMOKE; (Houyoux et al., 2000)). Biogenic emissions are generated with the

Biogenic Emissions Inventory System (BEIS) v3.61 (Pierce et al., 2002). Prescribed burning and

wildfire emissions are computed using the SmartFire 2 (Raffuse et al., 2009). Mobile emissions

are produced from the EPA's MOtor Vehicle Emission Simulator (MOVES) 2014a (US EPA,

MOVES2014a). The gas-phase chemical mechanism is CB05, and the aerosol module is AERO6.

Appel et al. (2013; 2017) provide details on the calculation of total $PM_{2.5}$ mass and speciated

aerosol mass, as well as model evaluation.

**2.3   Offline AOD calculation**

We calculate hourly AOD from the CMAQ model ($AOD_{CMAQ}$) offline from the archived

hourly, three-dimensional, speciated aerosol (i.e. sulfate, nitrate, ammonium, black carbon,

organic carbon, sea salt, soil dust) distribution and meteorological fields (i.e. relative humidity,

thereafter RH) using the Flexible Aerosol Optical Depth (FlexAOD) post-processing tool.

FlexAOD was originally developed to calculate aerosol optical properties for the GEOS-Chem

model. It is based on the NASA Codes for Computation of Bidirectional Reflectance of Flat

Particulate Layers and Rough Surfaces (Mishchenko et al., 1999). We adapt FlexAOD to CMAQ

by matching the aerosol speciation with GEOS-Chem based on Appel et al. (2013). Under the

assumption of spherical particles, aerosol optical properties are calculated based on Mie theory.

Given size distributions for each aerosol species, aerosol light extinction ($EXT_l$) at a given model

layer is calculated as follows (Curci, 2012):

$$EXT_l = \sum_{i=1}^{N} \frac{3}{4} \frac{\overline{Q_{e,dry,i} f_{RH_l,i}}}{r_{e,dry,i} \rho_i} M_{i,l} \qquad (4)$$





where $i$ refers to the species, $N$ is the number of aerosol species ($N = 5$: sulfate-nitrate-ammonium (SNA), OC, black carbon (BC), dust, sea salt), $\overline{Q_{e,dry,i}}$ is the Mie extinction efficiency of species $i$ averaged over the dry size distribution, $f_{RH_l,i}$ is the hygroscopic growth factor of species $i$ at given $RH_l$, $\rho_i$ is aerosol density of species $i$, $M_{i,l}$ is the aerosol mass of species $i$ at layer $l$, and

$r_{e,dry,i}$ is the dry effective radius. $AOD_{CMAQ}$ is then calculated as the vertical integral of $EXT_l$ across all model layers:

$$AOD_{CMAQ} = \int_{l=1}^{L} EXT_l \, dz \tag{5}$$

We use the recommended values of Drury et al. (2010) for aerosol density. The refractive index ($m$) in the default run is adapted from the Optical Properties of Aerosols and Clouds (OPAC)

database (Hess et al., 1998). As CMAQ does not explicitly simulate the size distribution of aerosols, we assume log-normal distributions for all species except for dust (assumed to be a gamma distribution). The effective radius ($r_e$), or the area-weighted mean radius of log-normal size distribution can be derived as:

$$r_{e,dry,i} = r_0 e^{(\frac{5}{2}ln^2\sigma_g)} \tag{6}$$

where $r_0$ is the specific modal radius, $\sigma_g$ is the geometric standard deviation. For the aerosol size distribution and density, we follow the recommended values of Drury et al. (2010) in the default run. We apply the single parameter $\kappa$ to represent the hygroscopic growth of SNA and organic carbon, which is developed by Petters and Kreidenweis (2007) based on the $\kappa$-Kohler theory, and it is the most commonly accepted function in the literature (Brock et al., 2016; Snider 2016). The

hygroscopic growth factor can be simplified as a function of parameter $\kappa$ and RH (Snider et al., 2016):

$$f(RH) = (1 + \kappa \frac{RH}{100-RH})^{1/3} \tag{7}$$



Koehler et al. (2006) suggest κ for SNA ($\kappa_{SNA}$) ranges from 0.33 to 0.72, with a mean of 0.53. The

hygroscopic growth factor of organic carbon ($\kappa_{oc}$) varies with species and is correlated with the

age of organics (Duplissy et al., 2011). Duplissy et al. (2011) and Jimenez et al. (2009) suggest κ

for organic carbon typically ranges from 0 to 0.2. We apply $\kappa_{SNA} = 0.53$, and $\kappa_{oc} = 0.1$ to represent

5     the hygroscopic growths of SNA and OC. For black carbon and sea salt, we apply the hygroscopic

growth factors reported in Chin et al. (2002). In addition to the default values, we test the

sensitivities of the derived $PM_{2.5}$ to uncertainties in aerosol optical property parameterization by

varying each parameter using values reported in the literature, as specified in Table 1.

## 2.4    Ground-based observations

10        AErosol RObotic NETwork (AERONET) is a federated instrument network that provides

ground-based information about aerosols including AOD, which is derived from sun photometer

measurements of direct solar radiation (Holben et al., 1998). We use Level-2 (cloud screened and

quality assured) daily average data from 13 sites over the Northeast USA. We also include

observed AOD from the Distributed Regional Aerosol Gridded Observation Networks

15    (DRAGON)-USA 2011 field campaign, co-located with the DISCOVER-AQ aircraft campaign.

The DRAGON campaign provides extensive sun photometer measurements of AOD at 38 sites

along the flight path of DISCOVER-AQ from July 1 to August 15, 2011, which were incorporated

into the AERONET database. To allow direct comparison with $AOD_{MAIAC}$ and $AOD_{CMAQ}$,

AERONET AOD measurements at 0.44 µm and 0.675 µm were interpolated to 0.55 µm using the

20    Angstrom exponent (the first derivative of AOD with wavelength, on a logarithmic scale) provided

(0.44 - 0.675 µm).

We use ground-based measurements of daily 24-hour average $PM_{2.5}$ from 152 EPA Air

Quality System (AQS) sites. Of the 152 sites, 13 sites have AERONET sites within 10 km (about



the resolution of CMAQ). We consider these 13 sites as "co-located" and use them to evaluate uncertainties in modeled PM$_{2.5}$/AOD relationships. We also use AQS aerosol speciation data at 54 sites which include the Chemical Speciation Network (CSN) and the Interagency Monitoring of Protected Visual Environments (IMPROVE) visibility monitoring network.

To evaluate the modeled vertical profile of ambient RH, we use ground-based soundings from 6 radiosonde sites over the Northeast USA. Aggregated daily data at 0:00 and 12:00 UTC are acquired from NOAA's Integrated Global Radiosonde Archive (IGRA), and modeled vertical profiles are sampled concurrently with radiosonde observations. We use the RH data calculated from vapor pressure, saturation vapor pressure and ambient air pressure (Durre and Yin, 2008).

**2.5   NASA DISCOVER-AQ 2011 Field Campaign**

The NASA DISCOVER-AQ (Deriving Information on Surface conditions from Column and Vertically Resolved Observations Relevant for Air Quality) aircraft campaign over Baltimore-Washington, D.C. in July 2011 provides extensive, systematic, concurrent measurements of aerosol chemical, optical, and microscopic properties. The NASA P-3B aircraft performed 14
flights which include 247 profiles (typically ranging from 0.4 to 3.2 km) over six DRAGON sites during the Baltimore-Maryland campaign in July 2011 (Crumeyrolle et al., 2014). In this study, we use the simultaneous measurements of aerosol composition (SNA, OC, BC), scattering, absorption, and extinction coefficients at dry (RH<40%), ambient and wet (RH>80%) environments. To reduce the random uncertainties of individual observations and to allow direct
comparison with CMAQ and ground-observations, we aggregate the daily aircraft profiles horizontally to six locations corresponding to the six sites, and vertically to CMAQ model layers, and then sample CMAQ modeled values consistently with observations.



## 3 Results and Discussion

### 3.1 Deriving surface PM$_{2.5}$ from satellite observations

We derive satellite-based PM$_{2.5}$ (thereafter PM$_{2.5\_MAIAC}$) over the Northeast USA for 2011 by taking the product of daily average CMAQ modeled PM$_{2.5}$/AOD relationships

(PM$_{2.5\_CMAQ}$/AOD$_{CMAQ}$) with MAIAC AOD (AOD$_{MAIAC}$, Eq. (1)). The unconstrained PM$_{2.5}$ estimates are independent of surface observations. As PM$_{2.5\_MAIAC}$ is determined as the product of observed AOD$_{MAIAC}$ and modeled PM$_{2.5\_CMAQ}$/AOD$_{CMAQ}$, the spatial patterns of PM$_{2.5\_MAIAC}$ will be affected by the spatial variations of both AOD$_{MAIAC}$ and PM$_{2.5\_CMAQ}$/AOD$_{CMAQ}$. Fig. 1a) shows the summertime average (JJA) AOD$_{MAIAC}$ at 1km resolution overlaid with AERONET observed

AOD. While we find high AOD over some populated urban areas such as New York City (NYC), high AOD$_{MAIAC}$ is also found over central New York State (NYS), away from the major anthropogenic sources. In CMAQ, while high PM$_{2.5\_CMAQ}$ are located at regions with major anthropogenic sources such as NYC region, AOD$_{CMAQ}$ shows a latitudinal dependence with higher AOD in low latitudes, which is due to 1) relatively high emissions of aerosol and its precursors

from anthropogenic and biogenic sources over MD, PA and NYC; and 2) latitudinal variations of RH that affect aerosol hygroscopic growth. The modeled PM$_{2.5\_CMAQ}$/AOD$_{CMAQ}$ varies spatially with standard deviation (SD) being 45 μg/m$^3$ per unit of AOD, which is mainly driven by the spatial variations of PM$_{2.5\_CMAQ}$ (R = 0.86). We find the overall spatial pattern of satellite-derived PM$_{2.5}$ is more correlated with modeled PM$_{2.5\_CMAQ}$/AOD$_{CMAQ}$ (R = 0.97) than observed AOD$_{MAIAC}$

(R = 0.8), suggesting that the large-scale spatial variability is determined largely by the model rather than by the satellite observations at least under our framework for the Northeast USA in summer. At smaller scales, over which we assume the spatial variability of PM$_{2.5}$/AOD is



homogenous, incorporating fine-resolution satellite data reveals stronger spatial gradients (e.g., enhancements along industrial corridors) than $PM_{2.5\_CMAQ}$ (Fig. 1b).

Besides improving the spatial resolution, satellite-derived $PM_{2.5}$ can correct model summertime biases in $PM_{2.5}$. Evaluating model with observed AOD from AERONET and $PM_{2.5}$

from AQS, we find an overall underestimate in both $AOD_{CMAQ}$ (Fig. 1c; normalized mean bias (NMB) = -44%) and $PM_{2.5\_CMAQ}$ (Fig. 1d; NMB= -17%) in summer. These concurrent underestimates should be reduced by taking the ratio. We find $PM_{2.5\_CMAQ}/AOD_{CMAQ}$ is overall consistent with the observed $PM_{2.5}/AOD$ sampled at co-located AQS-AERONET sites (NMB = 0.9%). AOD distributions retrieved from MODIS ($AOD_{MAIAC}$) agree better with AERONET AOD than $AOD_{CMAQ}$

than $AOD_{CMAQ}$ (NMB = 5%, Fig. 1f), though we find small low biases at two sites in New York City and at most DRAGON sites over Maryland. Our derived distribution of $PM_{2.5\_MAIAC}$ is thus closer to $PM_{2.5}$ observed at AQS sites than $PM_{2.5\_CMAQ}$ (NMB = 4.7% vs. 44% for $PM_{2.5\_CMAQ}$, Fig. 1g). However, the $PM_{2.5\_MAIAC}$ distribution is wider than observed at AQS: the lowest 5% is 5 vs. 7 $\mu g/m^3$ for $PM_{2.5\_MAIAC}$ vs. AQS $PM_{2.5}$, and the highest 5% is 16 vs. 13 $\mu g/m^3$. We find that

$PM_{2.5\_MAIAC}$ is biased high over New York City, coastal regions of Massachusetts, on the borders of upstate New York, and northern Vermont. Evaluation of $PM_{2.5\_MAIAC}$ in other seasons show larger biases and uncertainties (Fig. S1). In the following sections, we will discuss in detail the sources of uncertainties and biases of satellite-derived $PM_{2.5}$. We quantify the uncertainties in terms of bias (systematic) and random uncertainty. The bias uncertainty is linked to the overall

accuracy, while the random uncertainty reflects random fluctuations in measurements or the imprecision of the model resulting from imperfect modeling assumptions and simplifications.





### 3.2 Evaluation of satellite observed AOD

$AOD_{MAIAC}$ in general agrees well with AERONET observations with spatial R= 0.83, temporal R = 0.85, MB = -0.01, and RMSE =0.07. The performance of $AOD_{MAIAC}$ evaluated at Northeast US AERONET sites is consistent with evaluation of Superczynski et al. (2017) over

North America (R = 0.82, MB = -0.008). We find, however, that $AOD_{MAIAC}$ in winter (December-January-February, DJF) is biased high by 49% (MB = +0.02) on average. The wintertime overestimate is likely due to residual snow contamination which is below the detection limit, even though we applied a stringent data quality filter to remove pixels flagged as snow. We find the wintertime overestimate is most evident over northern latitudes (e.g. AERONET sites in

Massachusetts, NMB ranges from 80% to 180%), where snow occurs more often. The NMB of $AOD_{MAIAC}$ are 15% in MAM, -5% in JJA, and 17% in SON respectively, though the quantile range of the error is large, suggesting that single observations have large random uncertainties (Fig. 2). Taking the 1σ standard deviation of the normalized biases as a metric of random uncertainty, we estimate the uncertainties of daily satellite observations to be around 80% in DJF, 60% in MAM

and SON, and 50% in JJA. Spatial and/or temporal averaging can reduce random errors of satellite observations, which is evidenced as the smaller spread of errors that for monthly averages at the same spatial resolution, or daily data at coarser (10km) resolution, but it does not reduce the overall MB between $AOD_{MAIAC}$ and $AOD_{AERONET}$ (Fig. 2). We find that spatially averaging $AOD_{MAIAC}$ to 10 km leads to an overall increase of $AOD_{MAIAC}$. Temporal averaging, on the other hand, leads to

an overall decrease in $AOD_{MAIAC}$ except for DJF, leading to smaller positive MB in SON (7%) and MAM (7%), while larger negative MB in JJA (-8%) and positive MB (67%) in DJF.



### 3.3 Evaluation of modeled PM$_{2.5}$/AOD relationships

Three factors contribute to the overall uncertainty in the modeled PM$_{2.5}$/AOD relationship: 1) PM$_{2.5\_CMAQ}$; 2) AOD$_{CMAQ}$; 3) the relation between 1) and 2). We evaluate uncertainties of the three factors at 13 paired AQS-AERONET sites (within 10 km of each other; about the resolution of CMAQ). Figure 3 shows the distribution of the biases of modeled daily PM$_{2.5\_CMAQ}$, AOD$_{CMAQ}$ and PM$_{2.5\_CMAQ}$/AOD$_{CMAQ}$ compared with observations. Generally, PM$_{2.5\_CMAQ}$ biases vary seasonally: from +42% in DJF to -39% in JJA on average. In contrast, AOD$_{CMAQ}$ biases show weaker seasonality. The normalized MBs of AOD$_{CMAQ}$ are 3% in DJF, -16% in MAM, -7% in JJA and -20% in SON. On the daily scale, biases of AOD$_{CMAQ}$ are weakly correlated with the biases of PM$_{2.5\_CMAQ}$ (R = 0.14), suggesting model biases in AOD do not necessarily reflect biases in modeled PM$_{2.5}$. This is in contrast with prior analysis at the global annual scale where emission biases drive similar biases in AOD and PM$_{2.5}$ (van Donkelaar et al., 2013). The better accuracy of emissions in the Northeast USA than elsewhere in the world allows processes other than emissions to be more important for the Northeast USA. We find the seasonal biases in modeled PM$_{2.5}$ are retained in the PM$_{2.5\_CMAQ}$/AOD$_{CMAQ}$ ratio, which is even larger than the biases of PM$_{2.5\_CMAQ}$ in DJF, MAM and SON. As both PM$_{2.5\_CMAQ}$ and AOD$_{CMAQ}$ are biased low in JJA, the modeled PM$_{2.5}$/AOD bias (-20%) is smaller than that of PM$_{2.5\_CMAQ}$ (-39%). Biases in PM$_{2.5\_CMAQ}$ and AOD$_{CMAQ}$ are oppositely signed in fall, leading to largest mean biases of modeled PM$_{2.5}$/AOD (+74%). The spread of the biases of PM$_{2.5\_CMAQ}$/AOD$_{CMAQ}$ is larger than that of PM$_{2.5\_CMAQ}$ and AOD$_{CMAQ}$, with the standard deviation ranging from 50% in JJA to 100% in SON.




### 3.4 Relative importance of satellite AOD vs. modeled PM$_{2.5}$/AOD to uncertainties in satellite-derived PM$_{2.5}$

We have shown that both satellite AOD and modeled PM$_{2.5}$/AOD are subject to large uncertainties at the daily time-scale. To directly compare the relative importance of the biases of

satellite AOD vs. model PM$_{2.5}$/AOD on the satellite-derived PM$_{2.5}$, we scale the biases of modeled PM$_{2.5}$/AOD with daily AOD$_{MAIAC}$, so that the biases are expressed as the unit of PM$_{2.5}$ (µg/m$^3$):

$$\Delta PM_{2.5\_AOD} = (AOD_{MAIAC} - AOD_{AERONET}) \times \frac{PM_{2.5\_CMAQ}}{AOD_{CMAQ}} \qquad (8)$$

We then scale the of biases of AOD$_{MAIAC}$ with daily modeled PM$_{2.5}$/AOD relationship:

$$\Delta PM_{2.5\_Rel} = \left( \frac{PM_{2.5\_CMAQ}}{AOD_{CMAQ}} - \frac{PM_{2.5\_AQS}}{AOD_{AERONET}} \right) \times AOD_{MAIAC} \qquad (9)$$

We can also interpret $\Delta PM_{2.5\_AOD}$ and $\Delta PM_{2.5\_Rel}$ as the changes in derived PM$_{2.5}$ if 'true' observed AOD or PM$_{2.5}$/AOD instead of AOD$_{MAIAC}$ or modeled PM$_{2.5}$/AOD. As shown in Fig. 4a, mean biases caused by modeled PM$_{2.5}$/AOD are +9.2 µg/m$^3$ in DJF, +2.8 µg/m$^3$ in MAM, -3.3 µg/m$^3$ in JJA, and +7.7 µg/m$^3$ in SON respectively, which introduces larger biases to the derived PM$_{2.5}$ than the MAIAC satellite product in all seasons (7.6 µg/m$^3$ in DJF, +1.3 µg/m$^3$ in MAM, -0.7 µg/m$^3$ in

JJA, and 0.9 µg/m$^3$ in SON). Overall, satellite AOD contributes an error (root mean squared $\Delta PM_{2.5\_AOD}$) of 8.3 µg/m$^3$ to daily satellite PM$_{2.5\_MAIAC}$ with smallest error in JJA (5.1 µg/m$^3$) and largest error in DJF (13.2 µg/m$^3$), while modeled PM$_{2.5}$/AOD contributes an error of 10.8 µg/m$^3$ (root mean squared $\Delta PM_{2.5\_Rel}$) with smallest error in JJA (6.5 µg/m$^3$) and largest error in SON (15.2 µg/m$^3$). The spread of the biases is larger for modeled PM$_{2.5}$/AOD than that for MAIAC

AOD except for DJF. Our findings are consistent with Ford and Heald (2016), who estimate two times larger uncertainties in modeled PM$_{2.5}$/AOD relationships than observations using a higher-resolution (nested) version of the GEOS-Chem model and MODIS Dark Target AOD (Collection 6).



At the daily time scale, both $\Delta PM_{2.5\_AOD}$ and $\Delta PM_{2.5\_Rel}$ show large day-to-day variability: the $1\sigma$ standard deviation is 10.5 $\mu g/m^3$ for daily $\Delta PM_{2.5\_AOD}$ and 8.3 $\mu g/m^3$ or daily $\Delta PM_{2.5\_Rel}$. Next, we evaluate the dependence of the biases of satellite-derived $PM_{2.5}$ (denoted as $\Delta PM_{2.5\_MAIAC}$, evaluated with AQS observed $PM_{2.5}$) on $\Delta PM_{2.5\_Rel}$ versus $\Delta PM_{2.5\_AOD}$ by evaluating the Pearson

correlation coefficients (R). Overall, $\Delta PM_{2.5\_MAIAC}$ is more strongly correlated with $\Delta PM_{2.5\_Rel}$ (R = 0.85) than that with $\Delta PM_{2.5\_AOD}$ (R = 0.53), indicating the uncertainties of modeled $PM_{2.5}$/AOD are a more important driving factor to the uncertainties of daily satellite-derived $PM_{2.5}$, which could explain 72% variance ($R^2$) in $\Delta PM_{2.5\_MAIAC}$. In JJA, however, $\Delta PM_{2.5\_MAIAC}$ is moderately correlated with both $\Delta PM_{2.5\_AOD}$ (R = 0.48) and $\Delta PM_{2.5\_Rel}$ (R = 0.49), suggesting uncertainties of

modeled $PM_{2.5}$/AOD and satellite AOD contribute equally to the uncertainties of satellite-derived $PM_{2.5}$. We note that there is no statistically significant correlation between $\Delta PM_{2.5\_Rel}$ and $\Delta PM_{2.5\_AOD}$, with R ranging from -0.4 in SON to 0.23 in JJA, which suggests the errors caused by models and that caused by satellite AOD are independent of each other.

### 3.5    Factors leading to uncertainties in modeled $PM_{2.5}$/AOD relationship

Uncertainties in the modeled $PM_{2.5}$/AOD relationship mainly reflect uncertain aerosol speciation, aerosol vertical profiles, ambient RH, and parameterizations for aerosol optical properties including aerosol density, size distribution, refractive index and hygroscopic growth. Here we quantify the uncertainties from each factor and evaluate their impacts on the derived $PM_{2.5}$.

### 3.5.1    Aerosol speciation

Aerosol optical properties vary with chemical composition. Model biases in the aerosol composition also affect the overall particle hygroscopicity. For the same $PM_{2.5}$ abundance, variations in the aerosol composition may alter the particle optical properties especially hygroscopicity, and consequently the $PM_{2.5}$/AOD relationship. Fig. 5a compares the modeled



aerosol composition with ground-based observations averaged for each season. High biases in

$PM_{2.5\_CMAQ}$ in winter are largely due to model overestimates of OC by a factor of three, and low

biases in summer are due to a combination of underestimated SNA and OC. As a result, CMAQ

overestimates the fraction of OC by about 20% in DJF, 15% in MAM, and less than 10% in other

seasons, while underestimates the fraction of SNA by 5% to 20% in all seasons.

To estimate the impacts of model biases in aerosol speciation on $AOD_{CMAQ}$ and

$PM_{2.5\_MAIAC}$, we keep the total aerosol mass the same, and redistribute AOD ($AOD_{CMAQ\_ir}$) of each

species $i$ based on observed fraction of each species (i.e. SNA, OC, EC, soil dust; sea salt was

excluded due to the limited ground-based measurements and its negligible contribution):

$$AOD_{CMAQ\_ir} = \frac{PM_{i\_obs}}{PM_{TOT\_obs}} \times \frac{PM_{TOT\_CMAQ}}{PM_{i\_CMAQ}} \times AOD_{CMAQ\_i} \qquad (10)$$

where $PM_{TOT\_obs}$ and $PM_{TOT\_CMAQ}$ are the total aerosol mass from observations and CMAQ

respectively, which are reconstructed by summing up SNA, OC, EC and soil dust. Next, we

estimate the uncertainty due to speciation as the differences in derived $PM_{2.5\_MAIAC}$ ($\Delta PM_{2.5\_spe}$)

using the redistributed $AOD_{CMAQ\_ir}$ instead of the original $AOD_{CMAQ}$, shown in Fig. 5b. As SNA

generally has the largest mass extinction efficiency, a low bias in SNA leads to an overall

underestimate of $AOD_{CMAQ}$, and therefore an overestimate of $PM_{2.5\_MAIAC}$, which is largest in

winter (MB = 2.2 µg/m$^3$, SD = 2.6 µg/m$^3$) and smallest in summer (MB = 0.7 µg/m$^3$, SD = 3.0

µg/m$^3$). The estimated biases due to speciation show similar seasonal cycles as the modeled

$PM_{2.5}$/AOD biases (Fig. 3c), suggesting that aerosol speciation errors contribute to the seasonality

in modeled $PM_{2.5}$/AOD biases. Overall, model-observation discrepancy in speciation causes an

error (root mean squared $\Delta PM_{2.5\_spe}$) of 4.0 µg/m$^3$. On a daily basis, the correlation (R) between

$\Delta PM_{2.5\_spe}$ and $\Delta PM_{2.5\_MAIAC}$ is over 0.5 for all seasons except JJA, which means model biases in

speciation alone can explain more than 25% variance ($R^2$) in $\Delta PM_{2.5\_MAIAC}$. Biases in speciation





in JJA have relatively smaller impacts on the derived $PM_{2.5}$, which contribute less than 1 μg/m$^3$

MB and shows weak correlation with $\Delta PM_{2.5\_MAIAC}$ (R = 0.15).

### 3.5.2 Aerosol vertical profile

A caveat on the results in the Sect. 3.5.1 is that we assume the model errors in speciation

are constant across all vertical layers, as AQS sites only provide observations near the surface. The

DISCOVER-AQ aircraft campaign measured vertical variations in aerosol composition, although

spatial and temporal coverage is limited. Figure 6a compares the modeled and observed vertical

distributions of SNA, OC and BC averaged over the DISCOVER-AQ campaign. We do not discuss

sea salt and dust here since they contribute a negligible portion of the total aerosol mass in this

region. Both model and observations show SNA contributes more than half of the total aerosol

across all vertical layers (Fig. 6). Aircraft observations show SNA decreases gradually with altitude

with a nearly constant vertical gradient, while $SNA_{CMAQ}$ is well mixed below 1.5 km, and starts to

decline at the same rate as $SNA_{aircraft}$ above 1.5 km (Fig. 6). CMAQ underestimates SNA below

1.5 km, but overestimates SNA at higher altitudes. The positive model bias of SNA at higher

altitudes may be due to excessive vertical transport, or overestimation of RH (Sect. 3.4.3) and

consequently overestimation of $SO_2$ oxidation rate. OC, on the other hand, is biased low at all

altitudes, which is likely due to inaccurate treatments in the production of secondary organic

aerosol (Zhang et al., 2009). The newer version of CMAQv5.1 shows higher SOA concentration

in summer with the introduction of new SOA species (Appel et al., 2017). BC is generally low

during the campaign (typically lower than 0.3 μg/m$^3$). $BC_{CMAQ}$ generally agrees well with $BC_{aircraft}$,

though $BC_{CMAQ}$ tends to overestimate BC between 1 km and 3 km. Figure 6b compares CMAQ

modeled and observed total aerosol mass (SNA + OC + BC) averaged during the campaign.



CMAQ modeled aerosol mass is on average biased low below 2 km, and biased high at higher

altitudes (Fig. 6b).

Next, we evaluate how the vertical distribution of aerosols relates to extinction. Figure 6c

compares the modeled and observed average vertical extinction profiles. We find, consistent with

the biases in mass, a low bias in the modeled extinction profile below 2 km, and high bias above

(Fig. 6c). The biases in extinction and the biases in mass have the same signs for more than 80%

of data pairs, are strongly correlated (R = 0.85). This suggests that the aerosol vertical profile of

extinction is mainly indicative of mass distribution. However, column AOD measures the vertical

integral of light extinction by aerosols, which means the modeled AOD biases would be

proportional to modeled surface $PM_{2.5}$ biases only if the biases in extinction are constant across all

vertical layers. Since the biases of extinction change sign at higher altitude, the AOD biases reflect

the competing effects of negative biases near the surface and positive biases at high altitudes,

which lead to an overall negative bias of $PM_{2.5}$/AOD relationship, consistent with the negative

NMB of $PM_{2.5}$/AOD in July shown in Fig. 3c.

To explore the causes of the model-observation discrepancy in extinction and the resulting

impacts on the satellite-derived surface $PM_{2.5}$, we calculate the vertical extinction profile in

CMAQ by replacing the modeled aerosol mass distribution (SNA, OC, BC), or total mass

extinction efficiency (MEE, total aerosol mass/extinction), or RH respectively with those of the

aircraft observations, as shown in Fig. 7a. Replacing the modeled aerosol mass with observations,

we find a decrease in extinction at high altitudes (above 2.5 km) and increase at low altitudes

(below 2.5 km), but replacing the aerosol mass alone does not explain all of the model-observation

differences. At high altitudes, only replacing the modeled total mass extinction efficiency without

changing the mass captures the observed extinction. We attribute the model overestimate of





extinction to model overestimation of extinction efficiency at high altitudes. A major contributor

to the model overestimate of total MEE is its excessive RH at high altitudes, which leads to an

overestimate of the hygroscopic growth. Replacing RH with observations largely corrects the high

biases aloft, but does not correct the low biases below 2 km remain (Fig. 7a). At lower altitudes,

the model low biases are due to model underestimates of both aerosol mass and total MEE. Model

underestimates of MEE are likely due to: 1) model uncertainties of the optical properties; 2) other

aerosols or gases (e.g. $NO_2$, $O_3$) or liquid clouds that can scatter or absorb light.

Figure 7b shows the biases of $PM_{2.5\_MAIAC}$ due to model uncertainties in vertical profiles of

aerosol mass or MEE or RH, estimated by calculating the changes in $PM_{2.5}$ when we replace the

model vertical profiles with observations. Since the altitude of the aircraft observation ranges from

0.3 to 3.4 km, we use modeled values for the layers below 0.3 km and above 3.4 km while

attempting to minimize the discontinuity at both boundaries through vertical interpolation. As

SNA and OC contribute most to extinction, we also evaluate the biases of vertical profiles of SNA

and OC separately. We find that replacing modeled aerosol mass with observed mass leads to small

positive biases in $PM_{2.5\_MAIAC}$ (MB = 0.05 $\mu g/m^3$, SD = 4.3 $\mu g/m^3$), due to the combined effects

of negative biases from SNA (MB = -2.5 $\mu g/m^3$, SD = 4.7 $\mu g/m^3$) and positive biases from OC

(MB = +1.9 $\mu g/m^3$, SD = 4.3 $\mu g/m^3$).

We further separate the model-observation discrepancy in the vertical profiles as

differences in total column mass versus in vertical profile shape by 1) keeping the modeled vertical

distribution but adjusting the mass of each species uniformly so that the total column mass is equal

to observation; 2) keeping the total column mass the same as in the model but redistributing the

aerosol based on the observed vertical profiles. We find that redistributing the aerosol vertical

profile leads to a positive mean bias in $PM_{2.5\_MAIAC}$ (MB = 1.1 $\mu g/m^3$, SD = 4.9 $\mu g/m^3$), while the




model-observation discrepancy in column mass leads to a negative mean bias (MB = -0.6 µg/m³, SD = 3.6 µg/m³) (Fig. 7b). The positive biases in the profile shape are mainly attributed to model biases of the vertical profile of SNA (MB = 1.2 µg/m³, SD = 5.0 µg/m³), which shows a larger fraction of SNA at higher altitude where aerosol is less effective at scattering light due to lower

RH. The negative MB of column mass reflects a combination of negative biases of SNA (MB = -4.1 µg/m³, SD = 5.6 µg/m³) due to model overestimates of SNA column mass, and positive bias of OC (MB = 6.7 µg/m³, SD = 4.4 µg/m³) due to model underestimate of column mass of OC. Model biases in mass extinction efficiency lead to a small positive MB of 0.6 µg/m³.

Using the observed $PM_{2.5}$/AOD acquired from paired AQS-AERONET sites, we estimate

that model biases in modeled $PM_{2.5}$/AOD lead to a negative MB of -0.9 µg/m³ with large day-to-day variability (SD = 9.8 µg/m³) during the DISCOVER-AQ campaign, reflecting the model biases from different sources as discussed above. Next, we evaluate which factor drives the daily variability in the modeled $PM_{2.5}$/AOD biases the most by evaluating R value between the estimated biases in modeled $PM_{2.5}$/AOD vs. that attributed to individual factors. We find model bias in

aerosol mass is the most deterministic factor for the biases in modeled $PM_{2.5}$/AOD (R = 0.82, Fig. 7c). Model biases in aerosol mass can be due to either biases in column mass or vertical profile shape. We find model biases in modeled $PM_{2.5}$/AOD are more dependent on the biases in aerosol column mass (R = 0.79), instead of vertical profile shape. Model biases in mass extinction efficiency show moderate correlation with model biases of $PM_{2.5}$/AOD (R = 0.56). While model

uncertainties in RH lead to an overall negative bias (MB = -1.7 µg/m³, SD = 7.4 µg/m³) to $PM_{2.5\_MAIAC}$, they are negatively correlated with model biases of $PM_{2.5}$/AOD (R = -0.25).




### 3.5.3 RH

Figure 7 suggests model biases in RH contribute a negative bias to the derived $PM_{2.5\_MAIAC}$ during the DISCOVER-AQ aircraft campaign. Here we evaluate the impacts of modeled RH ($RH_{CMAQ}$) biases on derived $PM_{2.5}$ throughout the year using six atmospheric soundings over the

Northeast USA. We only assess the impacts of RH on the optical properties (i.e. hygroscopic growth) of aerosols. Comparing $RH_{CMAQ}$ with observed RH ($RH_{obs}$), $RH_{CMAQ}$ is overall biased high with the largest biases in winter. To evaluate the resulting impacts on $AOD_{CMAQ}$, we re-calculate the extinction using observed ambient RH from the soundings instead of $RH_{CMAQ}$ in Eq. (4), as shown in Fig. 8. Replacing $RH_{CMAQ}$ with $RH_{obs}$ decreases extinction by ~50% on average

from the surface to 5km in both JJA and DJF (black lines in Fig. 8a and b). As AOD is the vertical integral of extinction, the total area between $EXT_{sonde}$ and $EXT_{CMAQ}$ (gray shading in Fig. 8 a and b) indicates the differences in AOD due to differences in RH. The differences in RH below 3km in DJF, MAM and SON contribute more than 80% to the total differences in AOD. In JJA, the contribution from higher versus lower altitudes is similar, despite model biases of RH are small

below 2 km.

We evaluate how the model-observation discrepancy in RH affects the derived $PM_{2.5}$ by calculating the changes in $PM_{2.5\_MAIAC}$ ($\Delta PM_{2.5\_RH}$) if $EXT_{sonde}$ is used instead of $EXT_{CMAQ}$. As expected, model errors in RH lead to a negative bias in derived $PM_{2.5\_MAIAC}$ of 2 µg/m$^3$ on average (Fig. 8c). The negative biases in $PM_{2.5\_MAIAC}$ due to RH are largest in spring (-3.5 µg/m$^3$), and

smallest in summer (-1.6 µg/m$^3$). While we find a large spread of $\Delta PM_{2.5\_RH}$ that extends from -10 to 5 µg/m$^3$ (SD = 4.5 µg/m$^3$, RMSE = 3 µg/m$^3$), there is no significant correlation between $\Delta PM_{2.5\_RH}$ and $\Delta PM_{2.5\_MAIAC}$ (R = 0.18, evaluated at nearby sites within 10 km), suggesting the



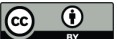

model uncertainty in RH is not a main contributor to the random uncertainties in satellite-derived $PM_{2.5}$.

### 3.5.4 Uncertainties in the parameterization of aerosol optical properties

In previous sections, we demonstrated that the satellite-derived $PM_{2.5}$ depends on the

accuracy of the model simulation. Even with a perfect simulation, satellite-derived $PM_{2.5}$ will be

sensitive to the parameterization of aerosol optical properties, which would affect the mass-

extinction efficiency. We evaluate the uncertainties associated with the parameterization of aerosol

optical properties by varying each parameter (Table 1), and calculate the corresponding changes

in the derived $PM_{2.5\_MAIAC}$. Figure 9 shows the range of uncertainty in annual average $PM_{2.5\_MAIAC}$

due to uncertain aerosol size distributions, hygroscopicity, refractive index and aerosol species

density.

The size of a particle is a defining characteristic of aerosol light extinction (Mishchenko et

al., 1999). To evaluate model sensitivities to the uncertainties in size distribution, we vary the $r_0$

of SNA from 0.05 to 0.15 with a 0.02 increase each time, to cover the range of values reported in

the literature. For OC, we calculate $AOD_{CMAQ}$ with $r_0 = 0.02$, 0.06, 0.09 and 0.12 μm, all values

used in previous studies (Hess et al., 1998; Chin et al., 2002; Highwood, 2009; Drury et al., 2010).

Annual average $PM_{2.5\_MAIAC}$ could vary by up to 5 μg/m$^3$ (32%) with the choice of modal radius

of either $r_{SNA}$ or $r_{OC}$, which is the largest source of uncertainty among the four parameters (Fig. 9).

We find that $AOD_{CMAQ}$ reaches a maximum with $r_{SNA} = 0.07$ μm ($r_{eff} = 0.12$ μm), and minimum

with $r_{SNA} = 0.05$ ($r_{eff} = 0.15$ μm), while $PM_{2.5\_MAIAC}$ reaches a maximum with $r_{SNA} = 0.05$ ($r_{eff} =$

0.09 μm), and minimum with $r_{SNA} = 0.11$ ($r_{eff} = 0.19$ μm), suggesting the impacts of size

distribution are nonlinear and non-uniform (Fig. S3). Mie scattering of a particle tends to be most

effective when the particle's diameter is near the wavelength of interest (0.55 μm). As hygroscopic




particle growth also affects the size distribution, depending on ambient RH and the hygroscopic growth factor, reducing (or increasing) the dry effective radius could either lead the bulk aerosol size closer to or further from 0.55 µm, and thus an increase or decrease in the extinction. For OC, as the effective radius and the hygroscopic growth factor are smaller than that of SNA, increasing

the modal radius leads to more effective scattering, thus larger $AOD_{CMAQ}$ and smaller $PM_{2.5\_MAIAC}$. Relative to the default $r_{OC} = 0.09$ µm as assumed in Drury et al. (2010), using the $r_{OC}$ (0.02 µm) recommended by Chin et al. (2002a) increases $PM_{2.5\_MAIAC}$ by 5 µg/m$^3$ (32%) on average, which would worsen the positive biases of $PM_{2.5\_MAIAC}$. Increasing $r_{OC}$ to 0.12 µm as recommended by Highwood et al. (2009) has little effect, decreasing $PM_{2.5\_MAIAC}$ by 2% on average.

The uncertainty of hygroscopicity lies in two aspects: the variations of the function shape and the uncertainties in the parameters. Figure S2 compares the function shape of κ function with the hygroscopic growth factors used by the IMPROVE network (Hand and Malm, 2006) and the default algorithm used for online calculation of AOD in CMAQ and that proposed by Chin et al. (2002) (Table 1). Using the DISCOVER-AQ aircraft data to evaluate the parameterization of

hygroscopic growth, we find that the κ parameter best characterizes the observed hygroscopic growth factor (Fig. S2c). Latimer and Martin (2018) similarly found that implementing a κ formulation instead of hygroscopic growth based on OPAC improved the GEOS-Chem representation of mass scattering efficiency. Thus, we choose the κ parameter to represent the hygroscopic growth factor, and the uncertainty estimate here only reflects the uncertainties with

parameter κ. In practice changes in aerosol composition could have even larger effects on hygroscopicity than uncertainties in κ as discussed in Sect. 3.5.1.

To test the uncertainties of κ parameter, we compute $AOD_{CMAQ}$ using the low (0.33) and high end of κ (0.72) for SNA as suggested by Koehler et al. (2006). As the hygroscopic properties



of inorganic salts are relatively well-known, the range of uncertainty for $f$(RH) of SNA is 30% at most (Fig. S2b). OC, on the other hand, is composed of thousands of species with distinct hygroscopicities. Assuming $\kappa_{OC}$ ranges from 0 (non-hygroscopic) to 0.2 (Jimenez et al., 2009; Duplissy et al., 2011), the range of $f$(RH) of OC can be as large as a factor of 2 at high RH>96%

(Fig. S2a). Despite the larger uncertainty of $\kappa_{oc}$, we find the overall impacts of the uncertainties of $\kappa_{oc}$ on the derived PM$_{2.5}$ (0.3 µg/m$^3$, 2% of annual average PM$_{2.5\_MAIAC}$) are smaller than that of $\kappa_{SNA}$ (1.6 µg/m$^3$, 11% of annual average PM$_{2.5\_MAIAC}$). The small impacts of $\kappa_{oc}$ reflect the relatively small portion and the less hygroscopic nature of OC. For single observations, varying $\kappa_{SNA}$ leads to a maximum increase in PM$_{2.5\_MAIAC}$ by 20% and a maximum decrease by 28%.

Varying $\kappa_{oc}$ increases PM$_{2.5\_MAIAC}$ by 10% or decrease PM$_{2.5\_MAIAC}$ by 18% at most. The overall impact of the uncertainties of $\kappa_{SNA}$ ranks the second among the four parameters for SNA, while $\kappa_{OC}$ has the smallest impacts on the derived PM$_{2.5}$ (Fig. 9).

The refractive index ($m$) determines the Mie extinction efficiency, which is subject to uncertainties mostly due to the lack of measurements (Kanakidou et al., 2005). $m_{SNA}$ in OPAC

(default value) is slightly different from that recommended in Chin et al. (2002) and Highwood (2009). Moise et al. (2015) suggest $m_{OC}$ varies by species, with its real part ranging from 1.37 to 1.65. We calculated additional AOD$_{CMAQ}$ by varying the real part of $m_{SNA}$ and $m_{OC}$ using the lowest and highest values reported in the literature. We find the annual average PM$_{2.5\_MAIAC}$ decreased by 0.8 µg/m$^3$ (6%) using the high end of $m_{R\_SNA}$, while increased by 1.3 µg/m$^3$ (9%) on average using

the low end. Though M$_{R\_OC}$ has a wider range of uncertainty, its impacts on PM$_{2.5\_MAIAC}$ (-4% to +6%) are smaller than that of M$_{R\_SNA}$. While the overall impacts on PM$_{2.5\_MAIAC}$ due to uncertainties of M$_{R\_SNA}$ are generally within 10% for single observations, PM$_{2.5\_MAIAC}$ can change by more than 20% under SNA dominated and high RH environments. The overall uncertainty due





to $M_{R\_OC}$ is generally within 5% for single observations, with a few cases (<10% of the total data) where the relative change in $PM_{2.5\_MAIAC}$ can exceed 10%.

As aerosol density ($\rho$) is assumed to be constant for each species, varying $\rho$ has the same effect on the extinction of given species. We vary the aerosol density of SNA from 1.65 to 1.83 g/cm$^3$ based on the uncertainty estimate from a laboratory study of Sarangi et al. (2016), which translates to an uncertainty of -3% to 7% for $AOD_{SNA}$, and the aerosol density of OC from 1.2 to 1.78 g/cm$^3$ following Park et al. (2006), which translates to an uncertainty in $AOD_{OC}$ ranging from -8% to 37%. We find aerosol species density, in general, contributes the smallest uncertainty to the satellite-derived $PM_{2.5}$. Varying $\rho_{oc}$ leads to annual average $PM_{2.5\_MAIAC}$ increased by 0.9 µg/m$^3$ (6%) and decreased by 0.6 µg/m$^3$ (3%) at most. As the aerosol density of inorganic salt is less uncertain, varying $\rho_{sulf}$ leads to negligible changes in annual average $PM_{2.5\_MAIAC}$ at both high (0.7 µg/m$^3$, 5%) and low ends (-0.5 µg/m$^3$, -2%).

## 4 Conclusions

We derive surface $PM_{2.5}$ distributions from satellite observations of AOD (MAIAC products) at 1 km resolution for 2011 over the Northeast USA using a geophysical approach that simulates the relationship between surface $PM_{2.5}$ and AOD with a regional air quality model (CMAQ) and offline AOD calculation package (FlexAOD). We find the fine spatial resolution of MAIAC AOD reveals more spatial details such as hot spots over populated urban areas or along major roadways. While the geophysical approach has shown promising values for mapping the $PM_{2.5}$ exposure at seasonal to annual scale (van Donkelaar et al., 2010; 2016), we show that estimating $PM_{2.5}$ from satellite AOD at the daily scale is not only subject to large measurement uncertainties of satellite observations, but more importantly, to uncertainty in daily variations of the relationship between surface $PM_{2.5}$ and column AOD. We take advantage of multi-platform *in*





*situ* observations available over the Northeast USA to quantify different sources of uncertainties in the satellite-derived $PM_{2.5}$, especially at the daily scale. We use observed AOD from the AERONET sun photometers to quantify uncertainties in satellite and modeled AOD; co-located AQS $PM_{2.5}$ and AERONET sites to evaluate modeled $PM_{2.5}$/AOD relationships; IMPROVE and

CSN aerosol speciation data to evaluate model uncertainties of aerosol composition; atmospheric soundings to evaluate modeled RH, as well as their impacts on $PM_{2.5}$ derivation. To assess the uncertainties associated with aerosol vertical profile, we use the extensive concurrent measurements of extinction and aerosol composition available from the NASA DISCOVER-AQ 2011 campaign over Baltimore-Washington, D.C. Finally, we estimate intrinsic uncertainties

associated with the model parameterization of optical properties, by testing sensitivities of satellite-derived $PM_{2.5}$ to variations in each parameter separately using FlexAOD.

As the relationship between surface $PM_{2.5}$ and column AOD is non-linear and spatiotemporally heterogeneous, satellite AOD alone is unable to fully resolve the spatial and temporal variability of ground-level $PM_{2.5}$. At the seasonal scale, we find large-scale spatial

variability of satellite-derived $PM_{2.5}$ is more correlated with the spatial variability in modeled $PM_{2.5}$/AOD than observed $AOD_{MAIAC}$. At the daily scale in the Northeast USA, modeled $PM_{2.5}$/AOD introduce larger mean biases to satellite-derived $PM_{2.5}$ than satellite retrievals, and uncertainties in modeled $PM_{2.5}$/AOD explain more than 70% variance in the uncertainties of satellite-derived $PM_{2.5}$, suggesting that the precision of daily satellite-derived $PM_{2.5}$ depends on

the capability of model to simulate the day-to-day variability of the relationship between $PM_{2.5}$ and AOD.

Uncertainties in modeled $PM_{2.5}$/AOD relationships can be attributed to several factors, including uncertain model aerosol speciation, vertical profiles, RH, as well as the parameterization

of aerosol optical properties. We find that seasonally varying biases in modeled $PM_{2.5}$/AOD reflect biases in aerosol speciation, particularly OC, which is overestimated in the cold season, and underestimated by CMAQ in the warm season. Biases in aerosol composition in turn affect aerosol hygroscopicity. The CMAQ model generally overestimates RH, especially at high altitudes (> 2

km), which contributes to an overall negative bias to satellite-derived $PM_{2.5}$. Using concurrent measurements of vertical profiles of aerosol extinction and composition available from the DISCOVER-AQ 2011 aircraft campaign, we show that the aerosol extinction is indicative of mass distributions, but the biases of modeled extinction vary with altitude, meaning that the model biases in vertically integrated column AOD do not necessarily reflect model biases in surface $PM_{2.5}$.

We attribute uncertainties in modeled vertical profiles to various factors including column mass, profile shape, mass-extinction efficiency and RH. We find that model uncertainties of column mass and the mass-extinction efficiency drive the variability in modeled $PM_{2.5}$/AOD uncertainty, while RH and aerosol vertical profile shape contribute some systematic bias.

Even with a model that perfectly simulates the distribution of aerosols, calculating AOD is

subject to additional uncertainties in aerosol size distributions, hygroscopic growth factors refractive indices and aerosol density. Our uncertainty analysis involving a series of sensitivity tests in FlexAOD indicates that for SNA, the uncertainties in size distributions contribute most to uncertainty in the derived $PM_{2.5}$ (32%), followed by the hygroscopicity parameter $\kappa$ (11%), refractive index (9%), and aerosol density (5%). For OC, size distribution is also the largest source

of uncertainty in the derived $PM_{2.5}$ (32%). Despite the large uncertainty of the hygroscopicity of OC, its impact on the satellite-derived $PM_{2.5}$ is negligible (2%), even smaller than uncertainties associated with the refractive index and aerosol density (6% each).

Based on this uncertainty analysis, we identify opportunities and directions to develop the



applications of satellite-derived PM$_{2.5}$ using the geophysical approach, especially at finer spatial and temporal scales. Van Donkelaar et al. (2016) found that calibration with ground-based PM$_{2.5}$ measurements improves the performance of satellite-based PM$_{2.5}$ at the annual scale, although such calibration is more challenging at short time scales (van Donkelaar et al., 2012). As the

5    uncertainties of satellite-derived PM$_{2.5}$ are due to a combination of multiple factors, calibration targeting specific sources of uncertainty would help further refine the geophysical approach. Additional collocated measurements of both PM$_{2.5}$ and AOD would be valuable to further evaluate the relationship between surface PM$_{2.5}$ and satellite AOD (Snider et al. 2015). Routine measurements of aerosol vertical profiles would facilitate uncertainty attribution and likely lead to

10   improved models and thereby reduce the overall uncertainty in satellite-derived PM$_{2.5}$. Quantifying source-specific uncertainties would not only facilitate future model improvement, but more importantly, benefit applications of the satellite-derived PM$_{2.5}$ products to health studies.




**Author contribution**: XJ, AF, AD, and RM designed the experiments. GC developed the FlexAOD code for CMAQ. AL provided MAIAC AOD data. KC and MK conducted the CMAQ simulation. XJ carried out the data analysis and prepared the manuscript with contributions from all co-authors.

5   **Acknowledgement**: Support for this study was provided by New York State Energy Research and Development Authority (Grant number: 91268) and NASA Health and Air Quality Applied Sciences Team (HAQAST, Grant NNX16AQ20G). The authors thank Melanie Follette-Cook of NASA GSFC and Morgan State University for her help in obtaining the DISCOVER-AQ aircraft data. We acknowledge useful discussions with Yang Liu from Emory University and Dan

10   Westervelt from Lamont-Doherty Earth Observatory of Columbia University. Although this manuscript was reviewed internally, it does not necessarily reflect the views or policies of the New York State Department of Environmental Conservation.





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





# Tables

*Table 1*: Optical properties used to calculate AOD$_{CMAQ}$ in FlexAOD. Values in square brackets represent the range of uncertainties in given parameter, which are used for the sensitivity runs in FlexAOD to quantify their impacts on the satellite-derived PM$_{2.5}$.

| | Sulfate | OC | BC | Sea Salt | Dust |
|---|---|---|---|---|---|
| Modal radius [a] ($r_0$, µm) | 0.11[b] [0.05[c] ~ 0.15] | 0.09[b] [0.02[d] ~ 0.12[c]] | 0.02[b] | 0.40[b] | |
| Geometric standard deviation [a] ($\sigma\_g$) | 1.6[b] | 1.6[b] | 1.6[b] | 1.5[b] | |
| Aerosol density ($\rho$, g/cm$^3$) | 1.7[b] [1.65, 1.83][e] | 1.3[b] [1.2, 1.78][f] | 1.0[b] | 2.2[b] | |
| Refractive Index ($m$) at 550 nm | 1.53[g] [1.43[d], 1.6[c]] $- i0.006$[g] | 1.53[g] [1.37, 1.65][h] $- i0.008$ | $1.75 - i0.44$[g] | $1.5 - i10^{-8}$[g] | $1.53 - i0.0055$[g] |
| Hygroscopic growth factor ($f$) at RH = 90% | 1.77 [1.58, 1.96][i] | 1.24[j] [1.0[k], 1.41[j]] | 1.4[d] | 2.4[d] | 1.0[d] |
| | 1.8[d] | 1.6[d] | 1.4[d] | 2.4[d] | 1.0[d] |
| | 5.1[k] | 1.0[k] | 1.4[d] | 2.4[d] | 1.0[d] |

*Note*:

a. Assuming log-normal distribution for aerosol species except for dust. The effective radius is calculated as: $r_e = r_0 e^{(\frac{5}{2}\ln^2 \sigma_g)}$.
b. Drury et al. (2010).
10   c. Highwood et al. (2009).
d. Chin et al. (2002)
e. Sarangi et al., (2016)
f. Park et al., (2006)
g. OPAC (Hess et al., 1998)
15   h. Moise et al., (2015)
i. κ parameter (Petters and Kreidenweis, 2007). The hygroscopic factor ($f$) is calculated as: f(RH) = $(1 + \kappa \frac{RH}{100\text{-}RH})^{1/3}$ following Snider et al. (2016), where κ = 0.53 in the default run, κ = 0.33 for the low end, κ = 0.72 for the high end.
j. Calculated from κ parameter equation, where κ = 0.1 in the default run, κ = 0.2 for the high end
20   (Jimenez et al., 2009; Duplissy et al., 2011).
k. Empirical hygroscopic growth factors used by the revised IMPROVE algorithm (Hand and Malm, 2006) to calculate light extinction (http://vista.cira.colostate.edu/Improve/the-improve-algorithm/). The revised IMPROVE algorithm assumes no hygroscopic growth for OC.



# Figures

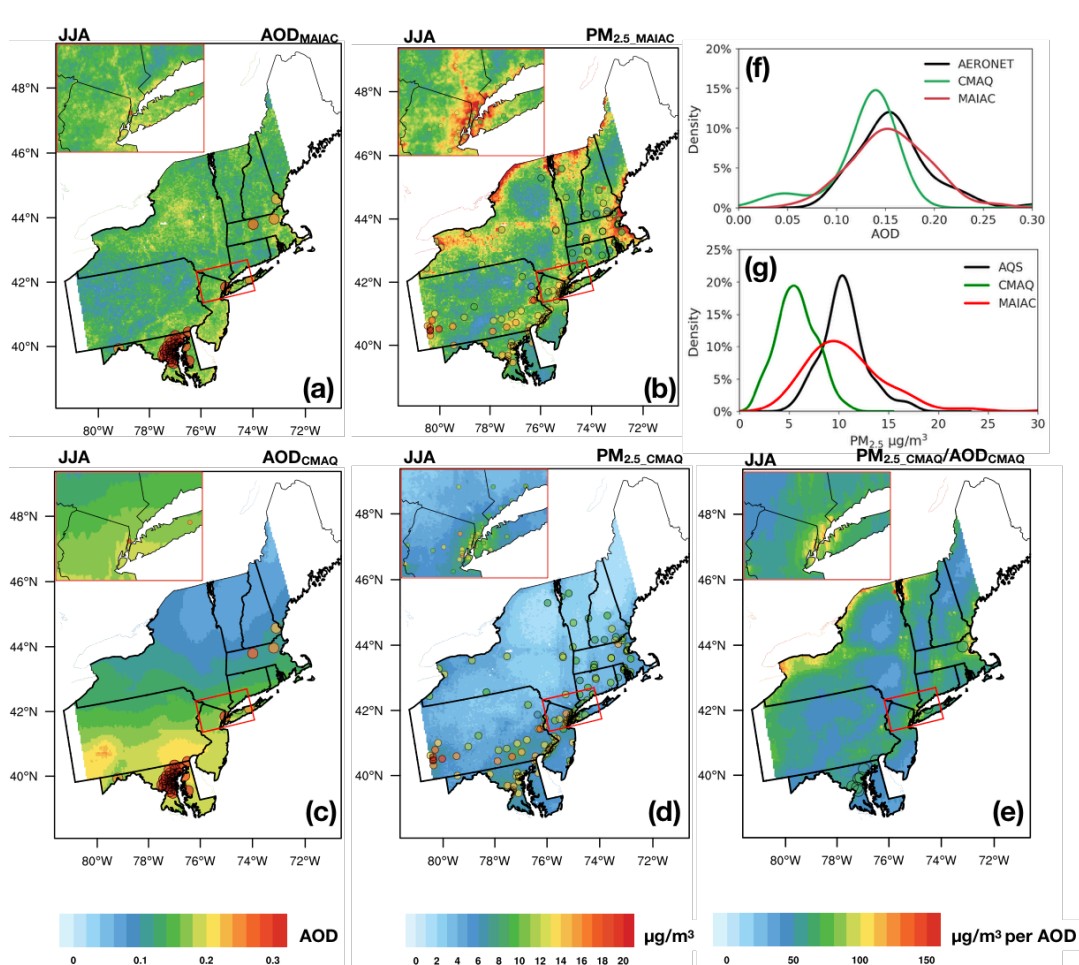

Figure 1 Summertime (JJA) average: (a) MAIAC AOD ($AOD_{MAIAC}$); (b) satellite-derived $PM_{2.5}$ ($PM_{2.5\_MAIAC}$); (c) CMAQ model AOD ($AOD_{CMAQ}$); (d) CMAQ model $PM_{2.5}$ ($PM_{2.5\_CMAQ}$); (e) CMAQ modeled $PM_{2.5}$/AOD ($PM_{2.5\_CMAQ}$/$AOD_{CMAQ}$) ratio overlaid with ground-based observations (AERONET, AQS, co-located AERONET and AQS sites) over the Northeast USA with zoom-in maps over the New York City region in the upper left corner. (f) Density plot of AOD showing the distribution of MAIAC, CMAQ and AERONET observed AOD sampled at AERONET sites. (g) Density plot of $PM_{2.5}$ showing the distribution of satellite-derived, CMAQ and AQS observed $PM_{2.5}$ sampled at AQS sites.




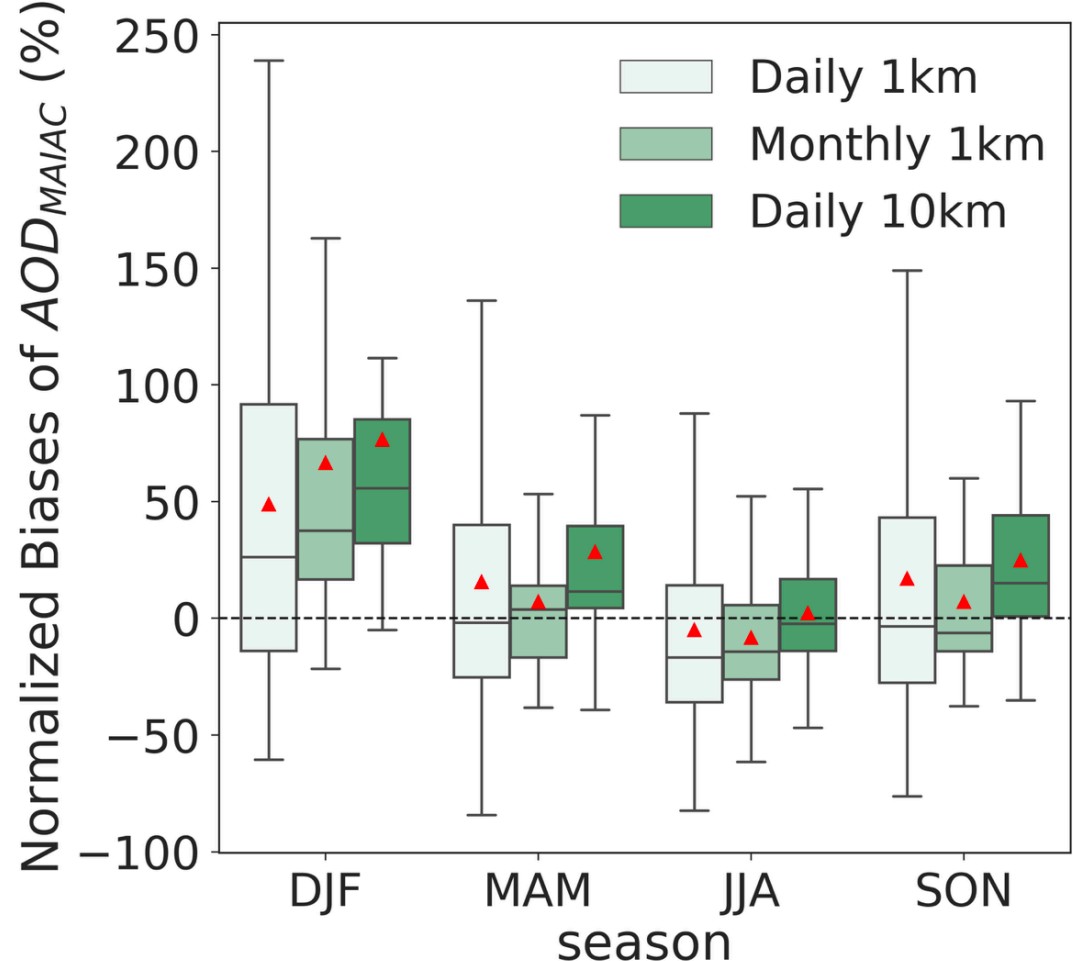

Figure 2 Distribution of normalized biases of AOD$_{MAIAC}$ evaluated at 52 AERONET (including DRAGON, only available for JJA) sites in four seasons of 2011 over the Northeast USA using daily MAIAC AOD at 1 km resolution, 10 km resolution, and monthly average MAIAC AOD composite (only including days when both satellite and AERONET measurements are available) at 1 km resolution. The box shows the quantile range (IQR) while the whiskers extend to show the rest of the distribution with outliers (points that are either 1.5×IQR or more above the third quantile or below the first quantile) removed. The red triangles show the seasonal mean normalized biases. Note that the normalized bias is an asymmetric metric, where model overestimates are unbounded whereas model underestimates are bounded by -100%, therefore the mean of normalized biases is typically higher than the median of the normalized biases.





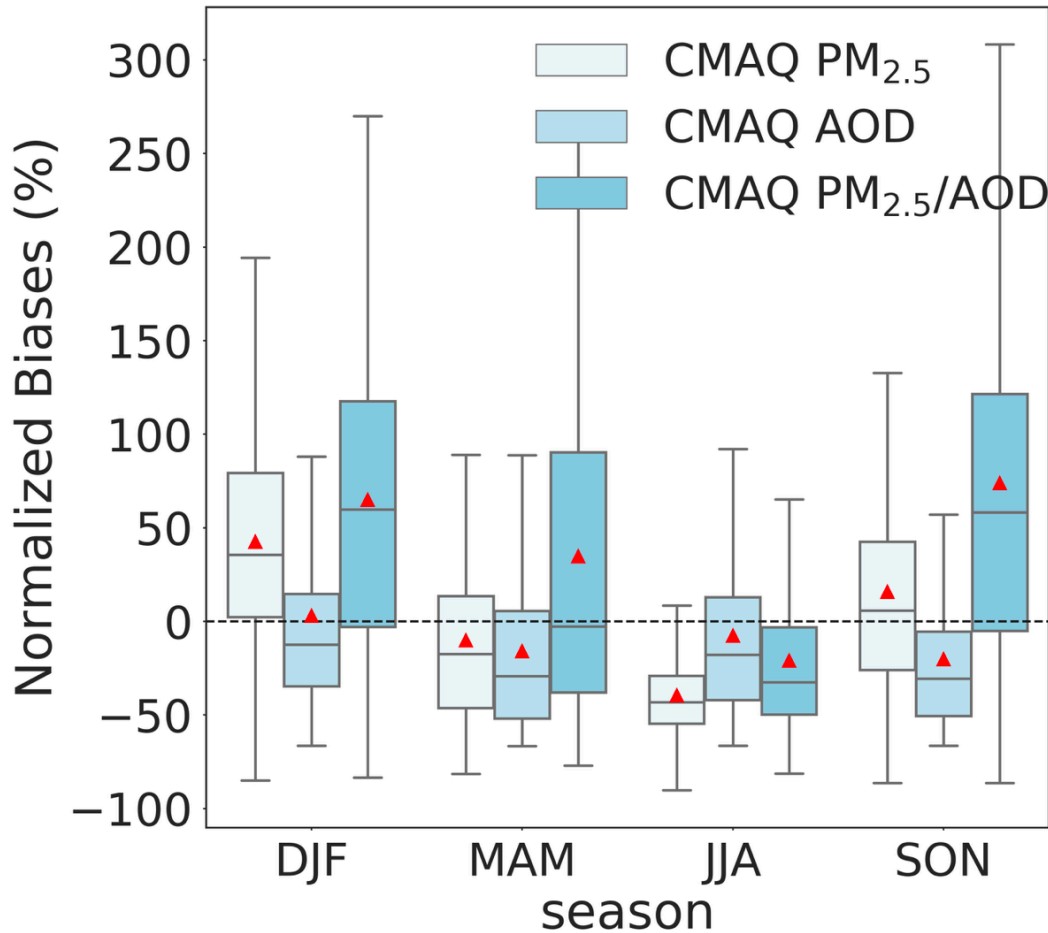

Figure 3 As in Figure 2 but for daily $PM_{2.5\_CMAQ}$, $AOD_{CMAQ}$, and $PM_{2.5\_CMAQ}/AOD_{CMAQ}$ in each season of 2011 evaluated at 11 co-located AQS-AERONET sites over the Northeast USA.





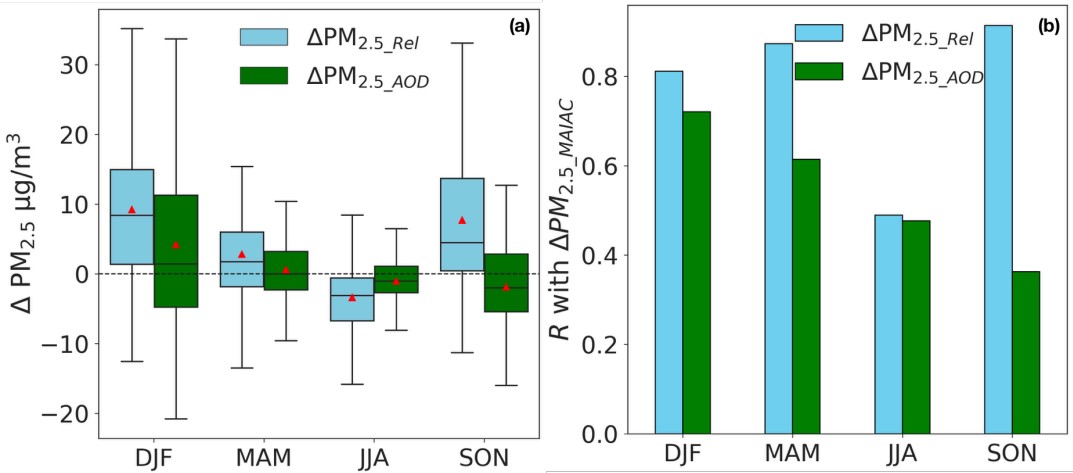

Figure 4 (a) Box plots comparing the distribution of biases in daily $PM_{2.5\_MAIAC}$ due to observational uncertainties in $AOD_{MAIAC}$ (green, $\Delta PM_{2.5\_AOD}$) versus model uncertainties in $PM_{2.5\_CMAQ}/AOD_{CMAQ}$ (blue, $\Delta PM_{2.5\_Rel}$), evaluated consistently at 11 co-located AQS-AERONET sites over the Northeast USA. (b) Pearson correlation coefficient between the biases in daily satellite-derived $PM_{2.5}$ ($\Delta PM_{2.5\_MAIAC}$, evaluated with AQS observations) and the biases in $PM_{2.5\_AOD}$ attributed to observational uncertainties in $AOD_{MAIAC}$ ($\Delta PM_{2.5\_AOD}$) versus model uncertainties in $PM_{2.5\_CMAQ}/AOD_{CMAQ}$ ($\Delta PM_{2.5\_Rel}$). $\Delta PM_{2.5\_AOD}$ is calculated by multiplying the biases of $AOD_{MAIAC}$ with daily modeled $PM_{2.5}/AOD$ relationships (Eq. (8)). $\Delta PM_{2.5\_Rel}$ is calculated by multiplying the modeled $PM_{2.5}/AOD$ biases with daily $AOD_{MAIAC}$ (Eq. (9)). The red triangles show the seasonal mean biases.





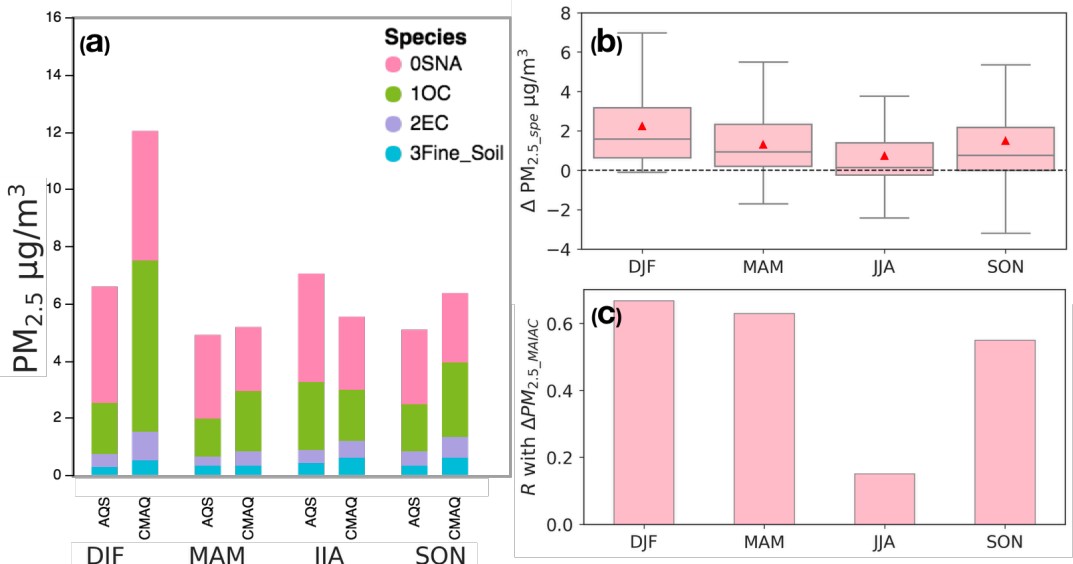

Figure 5 (a) Seasonal average $PM_{2.5}$ speciation from CMAQ vs. AQS observations in 2011 evaluated at 54 CSN and IMPROVE sites. (b) Box plots showing the distribution of estimated biases of daily satellite-derived $PM_{2.5}$ due to model biases in $PM_{2.5}$ speciation ($\Delta PM_{2.5\_spe}$) by season for 2011. Red triangles show the seasonal mean biases. (c) Pearson correlation coefficient between the biases in $PM_{2.5\_MAIAC}$ ($\Delta PM_{2.5\_MAIAC}$) and $\Delta PM_{2.5\_spe}$.





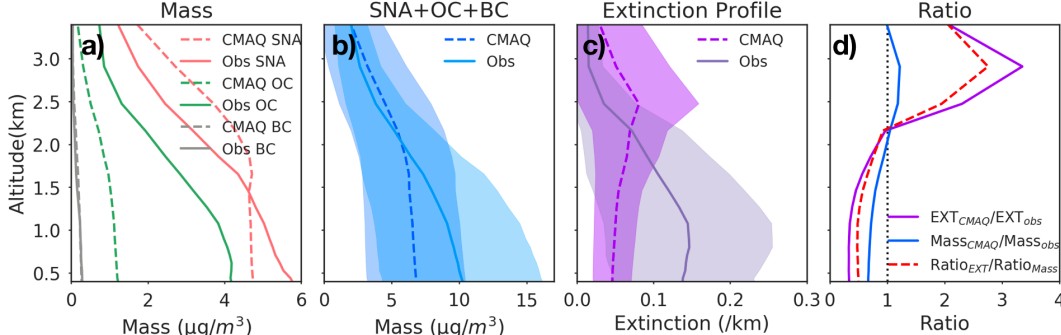

Figure 6 Campaign-mean vertical profiles of: (a) aerosol composition, (b) total mass
(SNA+OC+BC), and (c) extinction from CMAQ vs. observations from the DISCOVER-AQ 2011
Baltimore-Washington D.C. campaign. (d) Campaign-mean vertical profile of the model-to-
observation ratio of extinction (Ratio$_{EXT}$), total aerosol mass (Ratio$_{Mass}$) and Ratio$_{EXT}$/Ratio$_{Mass}$.
Aircraft observations are first aggregated to match model layers, and corresponding model values
are sampled concurrently with the time of observations. CMAQ modeled extinction is estimated
with FlexAOD using the default parameters in Table 1. The shading in (b) and (c) shows the
standard deviation of the day-to-day variability.



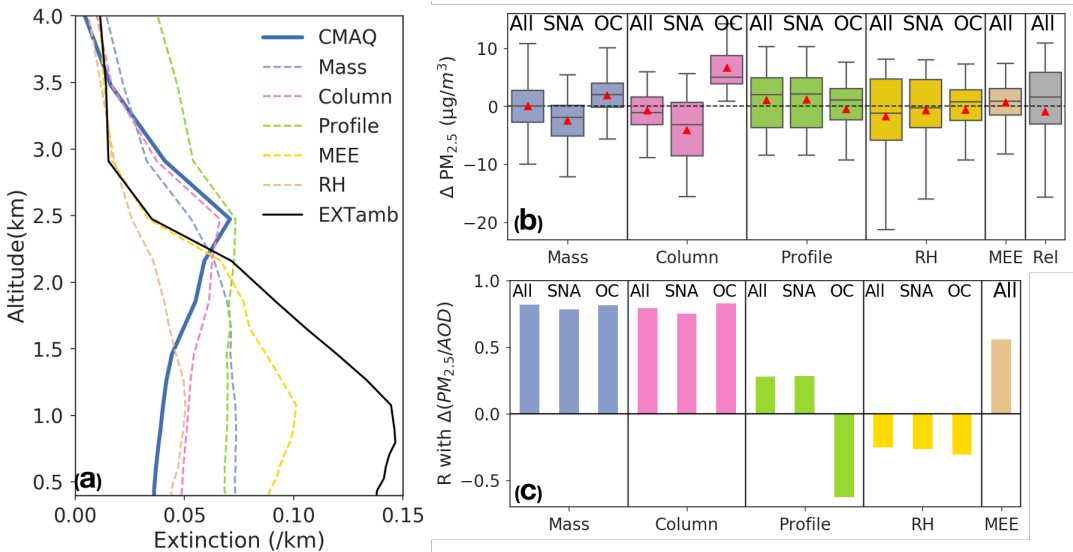

Figure 7 (a) Campaign-mean vertical profiles of extinction calculated from CMAQ speciated aerosol fields using FlexAOD, and that calculated by replacing modeled speciated aerosol mass (Mass), total column mass (Column), vertical profile shape (Profile), total mass extinction efficiency (MEE), relative humidity (RH) with that observed from DISCOVER-AQ 2011 Baltimore-Washington D.C. campaign. $EXT_{amb}$ is the aircraft observed vertical extinction profile. (b) Box plots showing the distribution of biases of $PM_{2.5\_MAIAC}$ attributed to each factor shown in (a), and the biases of $PM_{2.5\_MAIAC}$ attributed to modeled $PM_{2.5}/AOD$ (Rel). Red triangles show the mean biases. (c) Pearson correlation coefficient between the biases in modeled $PM_{2.5}/AOD$ relationships and the biases in modeled $PM_{2.5}/AOD$ attributed to individual factors shown in (b).



Figure 8 (a) DJF and JJA average vertical profiles of the CMAQ modeled vs. observed RH at 6 atmospheric soundings over the Northeast USA, and the modeled extinction vs. that calculated by replacing modeled RH with observed values. The gray area shows the difference in extinction two profiles, with the total area being the difference in AOD. (b) Box plots showing the impacts of model bias of RH on the derived $PM_{2.5\_MAIAC}$ ($\Delta PM_{2.5\_RH}$) in four seasons of 2011, which are calculated by comparing the $PM_{2.5\_MAIAC}$ minus the one calculated using observed RH. Red triangles show the mean biases.

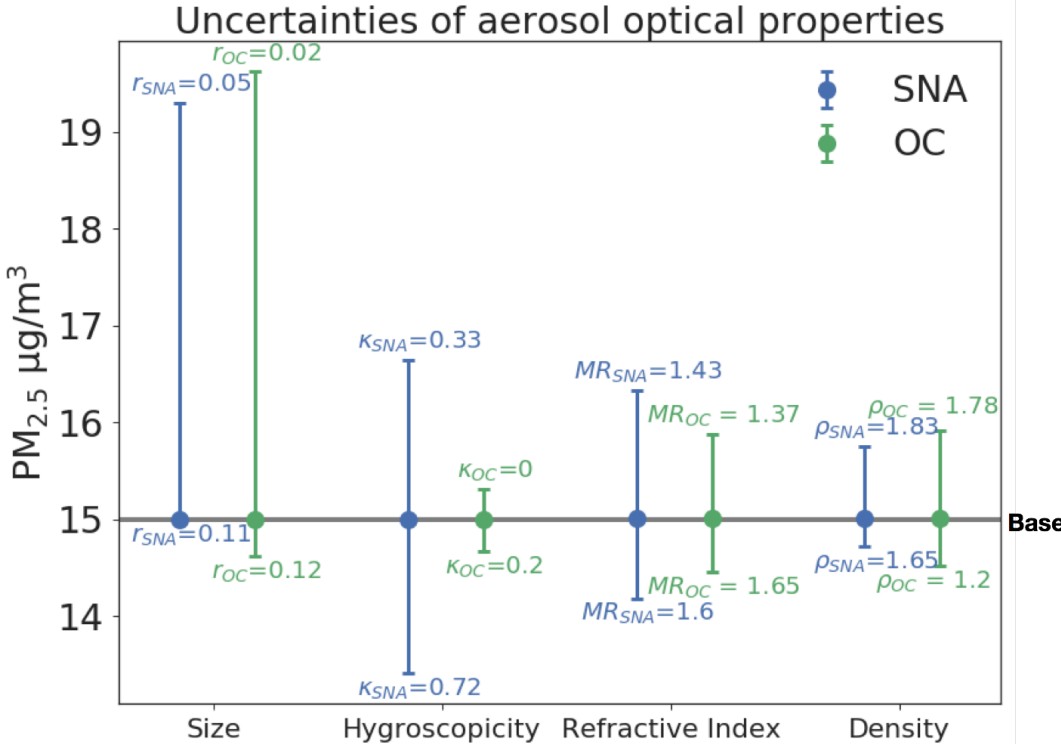

Figure 9 Uncertainties in annual average satellite-derived $PM_{2.5\_MAIAC}$ due to uncertainties of size
distribution, hygroscopicity, refractive index and aerosol species density of sulfate-nitrate-
ammonium (SNA; blue) and organic carbon (OC; green) sampled over AQS sites. The circle shows
the annual average satellite-derived $PM_{2.5\_MAIAC}$ using the default parameters to calculate
$AOD_{CMAQ}$ in FlexAOD (Table 1). The error bars represent the range of $PM_{2.5\_MAIAC}$ using different
values for each parameter. The labels indicate the corresponding minimum or maximum parameter
values that produce the range shown in $PM_{2.5\_MAIAC}$. The horizontal line at 15 µg/m³ indicates the
annual average $PM_{2.5\_MAIAC}$ calculated using default values for each aerosol optical property in the
base FlexAOD.