# Peer review of "Assessing uncertainties of a geophysical approach to estimate surface fine particulate matter distributions from satellite observed aerosol optical depth"

_Atmospheric Chemistry and Physics, 2018_

## Referee Comment (RC1) · Anonymous Referee #1 · 5 Nov 2018

This study evaluates the uncertainties associated with geophysical approaches to derive surface PM2.5, based on satellite AOD and modeled PM2.5/AOD. The authors go through a very detailed evaluation of all the potential factors, using ground-based observations of PM2.5, AOD, aircraft observations of aerosol extinctions/composition, and atmospheric soundings of RH over the Northeast United States. The analysis is very comprehensive, the paper is well written and I commend the authors for presenting the results in a succinct way on the figures.

One suggestion that I have for the authors is to present a figure with timeseries of

the daily variations in PM2.5, AOD, and PM2.5/AOD. The manuscript only contains barplots of the biases and pearson correlation coefficients, and there would be value for the reader to see the actual timeseries. I found Figure 1 very interesting in terms of displaying the contributions of different factors to spatial variability in satellite-derived PM2.5. Something similar to illustrate the controlling factors for the daily variability would be valuable.

---

## Referee Comment (RC2) · Anonymous Referee #2 · 11 Nov 2018

This is a very well written paper that explores sources of random and systematic bias on estimates of ground-level PM2.5 derived from satellite based AOD measurements and the ratio of AOD and PM2.5 from a regional air quality model.

The paper provides a review of the literature in this area, and then uses MODIS MA-IAC data and the CMAQ model to make PM2.5 estimates. Comparisons are made to Aeronet ground based measurements, and field measurements from the DISCOVER-AQ campaign. They carefully evaluate errors that originate from satellite AOD errors and from the modeled PM/5/AOD relationship.

[Figure]

The methodology, analysis, and data sources are all clearly described. The figures are well formulated and clear. I found the conclusions to be very clearly written and supported by the details in the manuscript.

There is one area where the authors should consider revisions. I think the hydroscopicity is an important element, and perhaps does not come across that way given that the details of the models for RH dependent particle growth are in supplementary material, and the statistics for RH are calculated like all the others. I would argue that factors like MEE and mass can be shown in box and whisker plots, but not the RH. The change of mass and extinction is very non-linear in RH. If the model says the RH is 90% and the field measurements say it is 60%, the situation is very different then if the model says RH is 60% and the observations say it is 30%. Can the analysis focus on the error due to RH errors that lead to substantial errors in the estimated aerosol growth - separate out high RH cases? This error source will be very seasonal and regional. Figure 8 hints at this, but the discussion still treats RH as if it is a factor that can be aggregated and treated like other linear factors, and I disagree.

---

## Author Comment (AC1) · 8 Dec 2018

**Reply to Reviewer 1**

**This study evaluates the uncertainties associated with geophysical approaches to derive surface PM$_{2.5}$, based on satellite AOD and modeled PM$_{2.5}$/AOD. The authors go through a very detailed evaluation of all the potential factors, using ground-based observations of PM$_{2.5}$, AOD, aircraft observations of aerosol extinctions/composition, and atmospheric soundings of RH over the Northeast United States. The analysis is very comprehensive, the paper is well written and I commend the authors for presenting the results in a succinct way on the figures.**

Reply: We would like to thank the reviewers for their time and effort to review our manuscript. We have revised the manuscript following the reviewers' suggestions.

**One suggestion that I have for the authors is to present a figure with time series of the daily variations in PM$_{2.5}$, AOD, and PM$_{2.5}$/AOD. The manuscript only contains barplots of the biases and Pearson correlation coefficients, and there would be value for the reader to see the actual timeseries. I found Figure 1 very interesting in terms of displaying the contributions of different factors to spatial variability in satellite-derived PM$_{2.5}$. Something similar to illustrate the controlling factors for the daily variability would be valuable.**

Reply:
That is a great point. We added a figure showing the temporal variability of regional average AOD$_{MAIAC}$, PM$_{2.5\_CMAQ}$/AOD$_{CMAQ}$, and PM$_{2.5\_MAIAC}$ (Figure 2). Similar to Figure 1, we show that the temporal variability of PM$_{2.5\_MAIAC}$ is mainly driven by the variability in PM$_{2.5\_CMAQ}$/AOD$_{CMAQ}$.

We have added the following discussions in the revised manuscript:

==The temporal variability in PM$_{2.5\_MAIAC}$ is also mainly driven by variability in PM$_{2.5\_CMAQ}$/AOD$_{CMAQ}$ (R = 0.61), with little temporal correlation between regional average AOD$_{MAIAC}$ and PM$_{2.5\_MAIAC}$ (R = 0.05, Fig. 2). At short time scales, the daily variability in regional average PM$_{2.5\_MAIAC}$ shows stronger correlation with PM$_{2.5\_CMAQ}$/AOD$_{CMAQ}$ in all seasons except for JJA, when PM$_{2.5\_MAIAC}$ are driven by variability in both AOD$_{MAIAC}$ (R = 0.5) and PM$_{2.5\_CMAQ}$/AOD$_{CMAQ}$ (R = 0.4, Fig. 2). Summertime AOD$_{MAIAC}$ is higher than wintertime AOD by 50%, while summertime PM$_{2.5\_MAIAC}$ is lower than in winter by 46%. Previous studies also found inconsistent seasonal cycles in AOD and PM$_{2.5}$ (Ford et al., 2013; Kim et al., 2015). We attribute the opposite seasonal cycle in PM$_{2.5\_MAIAC}$ and AOD$_{MAIAC}$ to three factors: 1) weak boundary layer ventilation in winter that leads to sharp vertical gradients of aerosol distribution (Kim et al., 2015); 2) higher RH in summer that leads to larger hygroscopic growth; 3) model==

overestimates of PM$_{2.5}$ (especially OC) in wintertime and underestimates of PM$_{2.5}$ in summertime, leading to an overestimate of the winter-to-summer decrease in PM$_{2.5\_CMAQ}$/AOD$_{CMAQ}$ (see section 3.3).

[Figure]

Figure 2 Regional 10-day running average of (a) MAIAC AOD (AOD$_{MAIAC}$, blue); (b) CMAQ modeled PM$_{2.5}$/AOD relationship (PM$_{2.5\_CMAQ}$/AOD$_{CMAQ}$, red); and (c) satellite derived PM$_{2.5}$ (PM$_{2.5\_MAIAC}$, green). The numbers on the upper left corner show the Pearson correlation coefficients (R) of PM$_{2.5\_MAIAC}$ with PM$_{2.5\_CMAQ}$/AOD$_{CMAQ}$ (red) and AOD$_{MAIAC}$ (blue).

**References:**

Ford, B. and Heald, C. L.: Aerosol loading in the Southeastern United States: reconciling surface and satellite observations, Atmos. Chem. Phys., 13(18), 9269–9283, doi:10.5194/acp-13-9269-2013, 2013.

Kim, P. S., Jacob, D. J., Fisher, J. A., Travis, K., Yu, K., Zhu, L., Yantosca, R. M., Sulprizio, M. P., Jimenez, J. L., Campuzano-Jost, P., Froyd, K. D., Liao, J., Hair, J. W., Fenn, M. A., Butler, C. F., Wagner, N. L., Gordon, T. D., Welti, A., Wennberg, P. O., Crounse, J. D., St Clair, J. M., Teng, A. P., Millet, D. B., Schwarz, J. P., Markovic, M. Z. and Perring, A. E.: Sources, seasonality, and trends of southeast US aerosol: an integrated analysis of surface, aircraft, and satellite observations with the GEOS-Chem chemical transport model, Atmos. Chem. Phys., 15(18), 10411–10433, doi:10.5194/acp-15-10411-2015, 2015.

---

## Author Comment (AC2) · 8 Dec 2018

**Reply to Reviewer 2**

**This is a very well written paper that explores sources of random and systematic bias on estimates of ground-level PM$_{2.5}$ derived from satellite based AOD measurements and the ratio of AOD and PM$_{2.5}$ from a regional air quality model.**

**The paper provides a review of the literature in this area, and then uses MODIS MAIAC data and the CMAQ model to make PM$_{2.5}$ estimates. Comparisons are made to Aeronet ground based measurements, and field measurements from the DISCOVER- AQ campaign. They carefully evaluate errors that originate from satellite AOD errors and from the modeled PM$_{2.5}$/AOD relationship.**

**The methodology, analysis, and data sources are all clearly described. The figures are well formulated and clear. I found the conclusions to be very clearly written and supported by the details in the manuscript.**

Reply: We would like to thank the reviewers for their time and effort to review our manuscript. We have revised the manuscript following the reviewers' suggestions.

**There is one area where the authors should consider revisions. I think the hygroscopicity is an important element, and perhaps does not come across that way given that the details of the models for RH dependent particle growth are in supplementary material, and the statistics for RH are calculated like all the others. I would argue that factors like MEE and mass can be shown in box and whisker plots, but not the RH. The change of mass and extinction is very non-linear in RH. If the model says the RH is 90% and the field measurements say it is 60%, the situation is very different then if the model says RH is 60% and the observations say it is 30%. Can the analysis focus on the error due to RH errors that lead to substantial errors in the estimated aerosol growth - separate out high RH cases? This error source will be very seasonal and regional. Figure 8 hints at this, but the discussion still treats RH as if it is a factor that can be aggregated and treated like other linear factors, and I disagree.**

Reply: That's a good point. We agree that RH errors should lead to larger uncertainties to satellite derived PM$_{2.5}$ at high RH. To address the reviewer's concern, we have added a figure showing the impacts of model bias of RH on the derived PM$_{2.5\_MAIAC}$ ($\Delta$PM$_{2.5\_RH}$) as a function of observed near-surface RH (Figure 9d). However, we'd like to argue that it is not possible to entirely separate out high RH cases because RH varies vertically, and the impacts of model biases of RH on the PM$_{2.5\_MAIAC}$ reflect the biases of RH integrated across all vertical layers. In Figure 9d, we use near surface radiosonde observations of RH (averaged over the first vertical layer in the model) to categorize the environment as humid or dry, with the limitation that it may not represent the conditions at higher altitudes.

We've added the following discussion in the revised manuscript:

The hygroscopic growth factor is nonlinearly correlated with RH, which increases more rapidly at high RH (> 80%) than at low to median RH (<80%, Fig. S1). Compared with median RH conditions, model RH errors lead to more than double $\Delta PM_{2.5\_RH}$ (-6.4 μg/m$^3$ versus 3 μg/m$^3$) when observed near-surface RH > 80% (Fig. 9d). At RH > 95%, we find that the $\Delta PM_{2.5\_RH}$ can be as large as -20 μg/m$^3$ (Fig. 9d).

[Figure]

Figure 9 (a) DJF and (b) JJA average vertical profiles of the CMAQ modeled vs. observed RH at 6 atmospheric soundings over the Northeast USA, and the modeled extinction vs. that calculated by replacing modeled RH with observed values. The gray area shows the difference in extinction two profiles, with the total area being the difference in AOD. (c) Box plots showing the impacts of model bias of RH on the derived $PM_{2.5\_MAIAC}$ ($\Delta PM_{2.5\_RH}$) in four seasons of 2011, which are calculated by comparing the $PM_{2.5\_MAIAC}$ minus the one calculated using observed RH. (d) Box plots show the influence of model RH biases on the derived $PM_{2.5\_MAIAC}$ ($\Delta PM_{2.5\_RH}$) as a function of observed near-surface RH.